

# Physical characterization of the boundary separating safe and unsafe AMOC overshoot behaviour

Aurora  Faure Ragani[1] and Henk A. Dijkstra[2,3]

[1]Institute for Mathematics, Utrecht University, Utrecht, the Netherlands
[2]Institute for Marine and Atmospheric research Utrecht, Department of Physics, Utrecht University, Utrecht, the Netherlands
[3]Center for Complex Systems Studies, Utrecht University, Utrecht, the Netherlands

**Correspondence:** Henk A. Dijkstra (h.a.dijkstra@uu.nl)

**Abstract.** The Atlantic Meridional Overturning Circulation (AMOC) is an important tipping element within the climate system as it may collapse due to a changing surface buoyancy forcing. Under scenarios of future greenhouse gas emission reductions, it has been suggested that the AMOC may undergo a safe overshoot. However, this was based on a rather conceptual model limiting the physical characterization of the boundary between safe and unsafe AMOC overshoot behaviour. Here, using a

fully-implicit global ocean model, we investigate the AMOC overshoot behaviour under different piecewise linear transient freshwater forcing scenarios. We clarify the physics of the collapse and recovery behaviour of the AMOC and show that the potential for a safe overshoot is tightly linked to a delicate balance of salt fluxes in the North Atlantic. More specifically, the sign of the time derivative of the integrated salt content in the northern North Atlantic is identified as an adequate indicator of the type of AMOC overshoot behaviour. The insights gained are relevant to inform climate policy strategies regarding emission

reductions, highlighting the necessity for thoughtful scenarios to prevent an AMOC collapse.

## 1   Introduction

A key component of the global ocean circulation is the Atlantic Meridional Overturning Circulation (AMOC), which plays an important role in shaping the climate of the Northern Hemisphere. The AMOC consists of the northward transport of warm surface waters and the southward return flow of colder, deep waters (Srokosz and Bryden, 2015; Buckley and Marshall,

2016). The AMOC is affected by density differences arising from heat and freshwater fluxes and has been identified as a potential tipping element in the climate system (Lenton et al., 2008). Models across the full range of complexity suggest that the AMOC could experience a rapid change of state in response to a gradual change in surface buoyancy forcing (Dijkstra, 2024), highlighting its susceptibility to significant shifts under evolving climate conditions.

The major process that can cause a rapid decline of the present-day AMOC is the salt-advection feedback. If the AMOC

weakens, less salt is transported northwards, decreasing the density in the north and further weakening the AMOC. In addition to the salt transport, an AMOC weakening also causes a decrease in northward heat transport. Yet, the damping times of salinity and temperature anomalies by the atmosphere are different: the atmosphere exerts quite a strong control on the sea surface temperature anomalies, but salinity in the ocean does not affect the freshwater flux. Hence, the positive salt-advection feedback can dominate over the negative heat-advection feedback (Marotzke, 2000).





The point beyond which a tipping element changes state is called a tipping point and can be characterized by the global warming level at which it occurs (Armstrong-McKay et al., 2022). Current climate change affects the forcing of the AMOC by making surface water warmer and less saline due to the addition of fresh water from melting ice – mainly from Greenland – and through increasing precipitation over the North Atlantic. Both of these forcing changes would decrease the meridional buoyancy gradient, weakening the AMOC (Gierz et al., 2015). The assessments of the tipping point thresholds have in part led 30 to the societal aspiration to restrict global warming to low levels such as $2.0°C$ or even $1.5°C$ above the pre-industrial period (UNFCCC, 2015). However, current emission levels and measured warming rates suggest that keeping the global warming within these restrictions will be difficult to achieve (Rogelj et al., 2023; Forster et al., 2024).

Our study is motivated by recent results where it is shown, using a conceptual box model, that global warming threshold may be temporarily exceeded without prompting a drastic change of the AMOC state (Ritchie et al., 2021). We consider this AMOC 35 overshoot problem using a fully-implicit global ocean model (described in Section 2) for which tipping points can be explicitly determined. This enables a detailed analysis of the underlying physical mechanisms that govern AMOC overshoot behaviour. In Section 3, we first determine the possible freshwater forcing trajectories that allow a safe overshoot of the tipping point varying rates of freshwater forcing as well as freshwater forcing peaks. Next, the analysis of the physics of the recovery and collapse is presented, where salt balances are monitored over different regions of the Atlantic basin. In Section 4, a reduced model of the 40 AMOC behaviour near the tipping point is studied to determine the precise boundary in parameter space separating safe from unsafe overshoot behaviour. Finally, in Section 5, we summarize and discuss the results.

## 2    Model and Methods

The description of the fully-implicit global ocean model used in this study is presented in Weijer et al. (2003) and Dijkstra (2007) to which the reader is referred for full details. In the AMOC model hierarchy (Dijkstra, 2024), this model is located 45 between idealised multi-basin ocean-only models and EMICs (Earth System Models of Intermediate Complexity). The model domain represents the global ocean, using continental geometry as well as bottom topography, with the longitude $\lambda$ ranging from $0°$ to $360°$ and the latitude $\theta$ from $85.5°S$ to $85.5°N$, on a $96 \times 38$ grid. The 12 vertical grid levels are non-equidistant with the most upper layer having a thickness of 50 m and the lowest of 1000 m. The model has no sea-ice component and the upper ocean is coupled to a simple energy-balance atmospheric model, in which only the heat transport is taken into account. Both 50 the neglected atmospheric moisture transport and sea-ice ocean interactions may affect the results below, but these effects are outside the scope of this study. The advantage of this model is that steady states and their stability can be determined without using any time stepping computation and explicit bifurcation diagrams can be efficiently computed (Wubs and Dijkstra, 2023).

The steady state solutions of the model versus parameters are computed as described in Dijkstra and Weijer (2005). Under Levitus restoring conditions for the surface salinity field, first a steady reference solution is determined for standard values 55 of the model parameters. From this solution, the freshwater flux that maintains the Levitus surface salinity field under steady state conditions, below referred to the Levitus flux $F_S^L$, is diagnosed. Moreover, this reference solution will be the starting solution for all the transient simulations in Section 3. Next, in addition to $F_S^L$, a freshwater perturbation is prescribed over a





region in the North Atlantic with domain $P = \{(\lambda, \theta) \in [300°\text{E}, 336°\text{E}] \times [54°\text{N}, 66°\text{N}]\}$. The strength of the perturbation is $\gamma_A F_S^P(\lambda, \theta)$, where $F_S^P(\lambda, \theta) = 1$ in the region $P$ and zero outside. The value of $\gamma_A$, expressed in Sv (Sverdrup, 1 Sv = $10^6$

$\text{m}^3\text{s}^{-1}$), controls the amplitude of the freshwater perturbation.

Thus, the total freshwater forcing prescribed is

$$F_S = F_S^L + \gamma_A F_S^P - Q, \tag{1}$$

where $Q$ is a compensation term determined such that

$$\int_{S_{oa}} F_S \cos\theta \, d\theta d\lambda = 0, \tag{2}$$

with $S_{oa}$ being the total ocean-atmosphere surface.

The meridional overturning stream function $\Psi$, expressed in Sv, is defined as the zonally-integrated (from $\lambda_W$ to $\lambda_E$) and vertically-accumulated meridional volume transport in depth and latitude coordinates:

$$\Psi(\theta, z) = r_0 \int_z^0 \int_{\lambda_W}^{\lambda_E} v(\lambda, \theta, z') \cos\theta \, d\lambda dz'. \tag{3}$$

where $r_0 = 6.378 \times 10^6$ m is the radius of the Earth. The bifurcation diagram (below) will show the maximum value of $\Psi$ in

the Atlantic basin below 500 m depth ($\Psi_A$) versus the control parameter $\gamma_A$, starting from the reference solution described above for $\gamma_A = 0$ Sv.

## 3    Results

The bifurcation diagram of the model is shown as the black dashed curve in Figure 1b. The reference solution has an AMOC strength of about 11 Sv (smaller than in observations due to the low resolution of the model (Dijkstra, 2007)) and with increas-

ing $\gamma_A$, the AMOC weakens. This branch of stable steady states ends at a saddle-node bifurcation, located at $\bar{\gamma}_A = 0.1855$ Sv, which will be referred below to as the tipping point. A branch of unstable steady states exists for decreasing $\gamma_A$, and leads to a second saddle-node bifurcation at $\gamma_A = 0.054$. With increasing $\gamma_A$, a stable branch of steady states exists for which the AMOC strength is near zero (and so not visible in Figure 1b). Thus, for a freshwater flux between $0.0522$ and $0.1855$ Sv, the AMOC is in a bi-stable regime, with a stable upper branch (representing the present-day AMOC) and a coexisting stable lower branch

(representing the 'off' (collapsed) AMOC state). Beyond the tipping point, only the lower branch exists. This bifurcation diagram will be the main reference to study the overshoot behaviour of the AMOC where a time-dependent freshwater forcing perturbation $\gamma_A = \gamma_A(t)$ (specified in later sections) will be applied.

### 3.1    Overshoot trajectories

Figure 1b also shows the AMOC response to two cases of transient forcing that overshoot the tipping point $\bar{\gamma}_A$. The freshwater

forcing grows linearly in time as shown in Figure 1a where the threshold is denoted by a horizontal line: in one case (blue





curve) the forcing goes beyond the tipping point after 900 model years. In the other case (red curve), the tipping point is
passed after 7400 model years. In Figure 1b, the AMOC trajectories of both forcing scenarios initially follow the stable steady
state branch. Since the forcing keeps increasing beyond the tipping point, the AMOC undergoes a change of state. The slower
forcing (red) causes the AMOC to stay closer to the steady state branch, whereas the faster forcing leads to a larger overshoot,
making the AMOC reach the off state for higher values of the parameter $\gamma_A$.

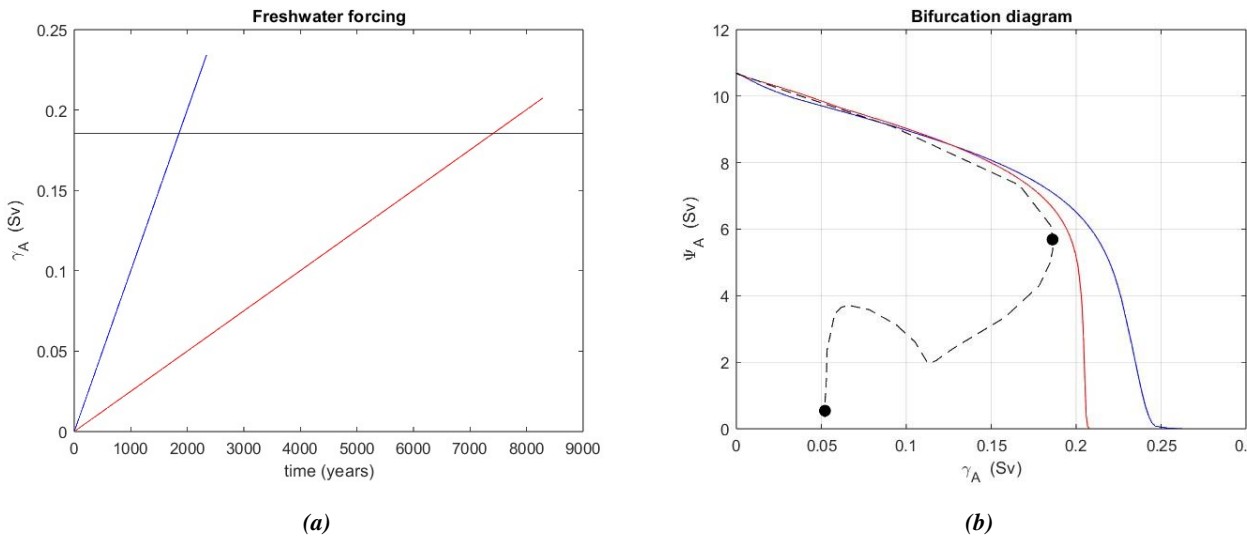

*(a)*                     *(b)*

**Figure 1.** (a) Two linear freshwater forcing trajectories with different slopes are shown as function of time; the horizontal line corresponds
to the tipping point $\bar{\gamma}_A = 0.1855$ Sv. (b) The AMOC strength curves associated to the forcings in (a) are plotted versus $\gamma_A$ in order to be
compared to the bifurcation diagram (dashed black curves). The black dots indicate the two saddle-node bifurcations.

The AMOC is considered a slow-onset tipping system, which means that crossing a tipping point threshold does not always
result in an immediate transition as seen in Figure 1b. This leaves the possibility that when the freshwater forcing is reversed,
the AMOC may undergo a safe overshoot, i.e., it does not collapse (Ritchie et al., 2021). To investigate safe versus unsafe
overshoots of the AMOC, we consider freshwater forcing scenarios $\gamma_A(t)$ as piecewise linear functions:

$$\gamma_A(t) = \begin{cases} m_1 t & 0 \leq t < t_1 \\ -m_2 t + h & t_1 \leq t < t_2 \\ -m_2 t_2 + h & t \geq t_2 \end{cases} \tag{4}$$

where $m_1, m_2 > 0$ and $h = (m_1 + m_2)t_1$ is such that $\gamma_A$ is continuous. Moreover, we chose $t_2$ such that the forcing after $t_2$
reaches a constant value, set to be half of the value of the tipping point $\bar{\gamma}_A$, i.e.,

$$t_2 = -\frac{1}{m_2}\left(\frac{\bar{\gamma}_A}{2} - h\right). \tag{5}$$





For $t < t_1$, the forcing mimics the current growth of the freshwater anomaly in the North Atlantic. To have an overshoot beyond
the tipping point, the parameters $m_1$ and $t_1$ will be chosen such that the maximum value of $\gamma_A$ (reached at $t = t_1$) is above the
tipping point $\bar{\gamma}_A$, hence

$$\gamma_A(t_1) = m_1 t_1 = -m_2 t_1 + h > \bar{\gamma}_A. \tag{6}$$

The subsequent linear decay tries to capture in the simplest way possible the decrease of the freshwater perturbation, which
could be faster or slower depending on the effort of society to lower the global emission rates of greenhouse gases.

We are going to consider three different properties of the freshwater perturbation applied in the region $P$ that can influence
the safe or unsafe overshoot of the AMOC: the peak, the rate of increase and the rate of decline. Case A, shown in Figure 2a,
represents two scenarios differing only in the rate at which the forcing decreases. Case B, presented in Figure 2c, features two
scenarios where only the rate of the forcing increase is varied. In case C instead (Figure 2e) we examine two scenarios where
the forcing has both different peaks and different decrease rates.

The two freshwater forcing trajectories of case A (Figure 2a) reach the same maximum value of $0.2384$ Sv but have different
decrease rates: $-1.0 \cdot 10^{-3}$ Sv/year for the blue scenario versus $-5.7 \cdot 10^{-4}$ Sv/year for the (dashed) green scenario. Since
both trajectories exceed the threshold value $\bar{\gamma}_A$, one would expect the AMOC to tip to the off state. However, as can be seen
in Figure 2b, the blue scenario results in a recovery and hence a safe overshoot, while the green one does not. In the safe
overshoot scenario, the AMOC spends a shorter time beyond the tipping point (131 years), enabling it to recover. Conversely,
in the unsafe overshoot, the AMOC remains beyond the tipping point for a longer time (167 years), and the slower decreasing
forcing causes a collapse.

As for Case B, Figure 2c shows two freshwater forcing trajectories that reach the same maximum value of $0.235$ Sv. However
the blue scenario has an initial rate of increase twice that of the green scenario. The forcing then decreases linearly with the
same rate in both scenarios. The AMOC behavior in Figure 2d shows that the slower increasing forcing causes a collapse, while
the faster increasing forcing allows a recovery. In both cases Figure 2b and Figure 2d, the AMOC is unable to recover when it
has a more prolonged exposure to forcing levels beyond the tipping point. This makes the time spent beyond the tipping point
an important factor that influences whether the AMOC collapses or recovers, as was also shown in Ritchie et al. (2021) using
a conceptual box model.

In Case C, both forcing trajectories (Figure 2e) have the same rate of increase but now they differ in terms of maximum
values of $\gamma_A(t)$. The blue scenario represents a smaller overshoot ($0.0495$ Sv beyond the tipping point) that peaks after 350
years, while the green scenario reaches a higher peak above the threshold ($0.0629$ Sv) after 370 years. They both get to the
same fixed level after 600 years, which means that the rate at which they decrease is different. In Figure 2f, it is clear that the
lower peak forcing leads to a weaker AMOC decrease and a subsequent recovery, while the higher forcing makes the AMOC
tip to the off state.





**Figure 2.** (a-b) Case A: (a) Freshwater forcing $\gamma_A(t)$. (b) AMOC strength trajectories. (c-d) Same for case B. (e-f) Same for case C. In the panels (b), (d) and (f), also the bifurcation diagram of the model is plotted (black dashed curves).



### 3.2 Salt Balances

In order to understand the physics underlying the results presented in Section 3.1, we consider the integral balance of salinity over the Atlantic basin, with meridional boundaries $S_\theta$ located at $35°S$ in the south and $60°N$ in the north. The overall salinity balance is given by (Dijkstra, 2007)

$$\Phi^b = \Phi^s - \Phi^a(\theta_s) + \Phi^a(\theta_n) - \Phi^d(\theta_s) + \Phi^d(\theta_n) + \frac{d}{dt} \int_{Atl} S \, dV \tag{7}$$

Here $\Phi^s$, given by

$$\Phi^s = - \int_{S_{oa}} S_0 F_S r_0^2 \cos\theta \, d\lambda d\theta \tag{8}$$

represents the (equivalent) salt flux through the ocean-atmosphere surface of the basin ($S_{oa}$), positive (negative) when evaporation is larger (smaller) than precipitation; $S_0 = 35$ psu indicates the reference salinity. The quantities $\Phi^a$ and $\Phi^d$ are the advective and diffusive salt fluxes through the boundary $S_\theta$ and given by

$$\Phi^a(\theta) = \int_{S_\theta} vSr_0 \cos\theta \, d\lambda dz \; ; \; \Phi^d(\theta) = - \int_{S_\theta} K_H \frac{\partial S}{\partial \theta} \cos\theta \, d\lambda dz, \tag{9}$$

where $K_H$ is the horizontal diffusivity. The last term on the right hand side of (7) measures the changes in time of the salt content stored in the Atlantic basin. This term will be zero for steady states, while it will play an important role in transient solutions. Finally, the residual $\Phi^b$ is used to monitor how well the salt balance is closed as in that case it should be zero. It was shown in Dijkstra and van Westen (2024) that the salt balance is closed in the steady case, i.e. along the bifurcation diagram in Figure 1b.

In the transient situation, we focus first on case A (Figure 2a-b), where a higher rate of forcing decline results in a recovering AMOC, while a slower decline rate weakens the AMOC until it collapses. Since both scenarios have forcing trajectories with identical peaks at the same time (355 model years), they behave in the same way during the first 355 years. Due to the different rates at which the forcing decreases, we will focus on this phase to characterize the safe and unsafe overshoots. In Figure 3a, three main contributions of the salt balance are considered: the surface flux $\Phi^s$, the tendency of the integrated salt content and the net salt flux $\Phi^{lat}$ through the lateral boundaries $S_{\theta_s}$ and $S_{\theta_n}$, where $\theta_s = 35°$S and $\theta_n = 60°$N. Here,

$$\Phi^{lat} = \Phi(\theta_s) - \Phi(\theta_n) \tag{10}$$

where $\Phi(\theta) = \Phi^a(\theta) - \Phi^d(\theta)$; $\Phi^{lat}$ is positive when salt is transported into the basin. Figure 3b shows the decomposition of the lateral fluxes into advective and diffusive fluxes at the northern and southern boundary of the Atlantic Ocean.

While $\gamma_A$ increases, the surface salt flux $\Phi^s$ decreases linearly (purple curve in Figure 3a) due to the input of freshwater in the Atlantic. However, this flux is still positive, which means that there is a net salt build up in the Atlantic basin that needs to be compensated. This compensation occurs through the lateral boundaries, where (note that $\Phi^{lat}$ is negative) salt is transported out of the basin. As expected, its absolute value is decreasing with increasing $\gamma_A$ and hence less salt is transported





out through the lateral boundaries. However, since the integrated salt content is decreasing ($\frac{d}{dt}\int_{Atl}SdV < 0$), the lateral salt

outflow is overcompensating the surface salt input. This implies that the lateral salt transport does not adjust quickly enough to the changing forcing, remaining stronger than needed for a steady balance. An important contribution to the lateral fluxes is due to $\Phi^a(\theta_s)$ (red curve in Figure 3b), which indicates the salt transported northwards at the southern boundary. As $\Phi^a(\theta_s)$ is negative, the salt is being transported out of the basin.

When $\gamma_A$ starts to decrease, $\Phi^s$ increases in both forcing scenarios, reflecting that less freshwater enters the ocean through

its surface. From this time onward, the dashed curves and the solid ones in Figure 3a begin to diverge. When $\gamma_A$ decreases more rapidly (solid curves), the increase of $\Phi^s$ is faster (solid purple curve). At the same time, $\Phi^{lat}$ (solid blue curve) reaches a minimum and then starts to increase again, indicating that the amount of salt transported out of the basin is reduced. These two factors make the Atlantic basin saltier as can be seen by the positive $\frac{d}{dt}\int_{Atl}SdV$ (solid green curve): less freshwater is put in and less salt is transported out. In the case of collapse (dashed curves), the combination of freshwater input and salt

transport outwards does not make the term $\frac{d}{dt}\int_{Atl}SdV$ positive: the salt flux through the lateral boundaries remains too strong to balance the slow increase of $\Phi^s$ and hence the salt storage in the Atlantic keeps decreasing.

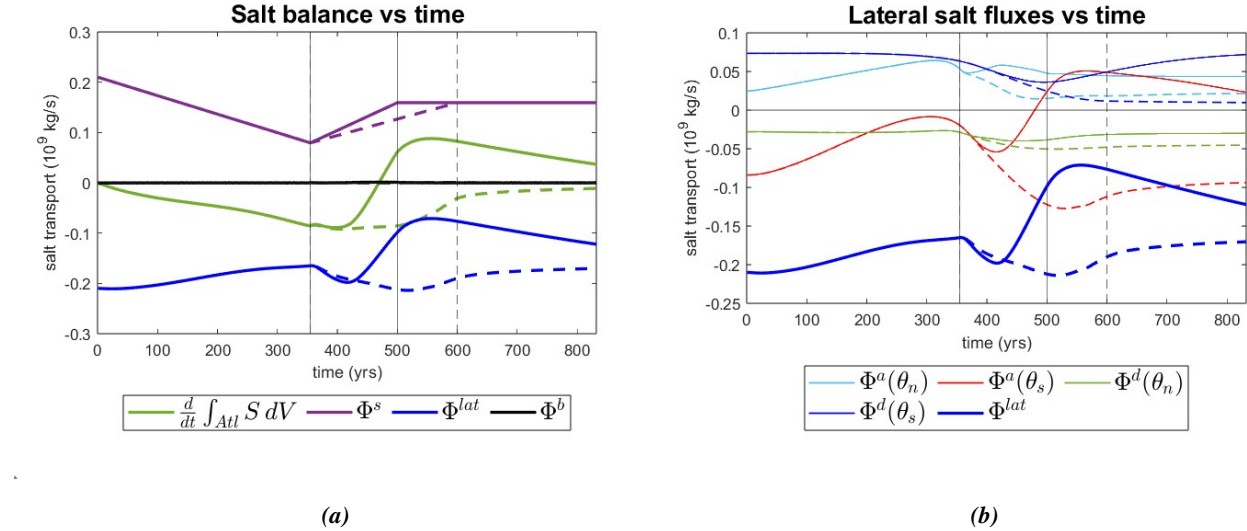

*(a)*                                                *(b)*

**Figure 3.** (a) Terms in the integrated salt balance and (b) Lateral salt fluxes over the Atlantic basin boundaries along the AMOC trajectories in Figure 2a-b. In both panels (a) and (b), the solid curves are for the recovery (safe overshoot) case, the dashed ones for the collapse (unsafe overshoot) case. The vertical lines mark the points when the forcing changes in time: at model year 355, the forcing reaches its peak in both cases; at the second vertical line (model year 500), the forcing in the safe overshoot scenario stops decreasing; and at the dashed vertical line (model year 600), the forcing in the unsafe overshoot scenario stops decreasing. In the legend of panel (b), $\theta_s = 35°\text{S}$ and $\theta_n = 60°\text{N}$.

To explain the different behaviour of the two scenarios in more detail, two boxes are defined: one in the North Atlantic and one in the South Atlantic. The northern box spans latitudes from $40°\text{N}$ to $60°\text{N}$ and the southern one extends from $15°\text{S}$ to $35°\text{S}$. Both are bounded zonally by the land bordering the Atlantic. We compute the box averaged densities $\rho_i$, salinities $S_i$





and temperatures $T_i$ $(i = s, n)$ by integrating over the total volume of each box. For case A, the meridional density difference $\Delta\rho := \rho_n - \rho_s$ is shown in Figure 4a together with the AMOC strength. The density difference behaves in the same way in the two scenarios up to the point where the freshwater forcing reaches its peak. Afterwards, the different decrease rates lead to changes in $\Delta\rho$. In the collapse case, $\Delta\rho$ keeps decreasing, while it has a minimum when the AMOC recovers after the overshoot. As shown in Figure 4b, $\Delta\rho$ is well correlated with the AMOC strength, consistent with other low-resolution ocean

model studies (Rahmstorf, 1996) and a consequence of the thermal wind balance. In the case of collapse, the density is plotted versus the AMOC strength for the whole time of the simulation, while in the recovery case, the relation is linear only during the increase of the forcing; in the period of the forcing decrease the AMOC has a more complex, time-dependent response (not shown).

We further decompose the density difference contribution by salinity ($\Delta S$) and temperature ($\Delta T$) meridional differences in

Figure 4d and Figure 4c. An increasing amount of freshwater is being introduced into the northern region of the Atlantic Ocean, which for $\gamma_A(t_0) = 0$ is more saline than the southern region ($\Delta S > 0$). This freshwater decreases the salinity difference ($\Delta S$) between the northern and southern boxes until it reaches 0 psu at around 300 model years. After that point until $t = t_1$ (when the forcing stops increasing) $\Delta S$ keeps decreasing, becoming more negative. The salinity in the north changes more than in the south (not shown). Not only in range (in the north, it spans a range between 0.65 psu and 0.85 psu depending on the scenario,

while only of around 0.1 psu in the south) but especially in shape. The main contribution to $\Delta S$ results from changes in the northern box. This is not surprising given the northern location of the prescribed freshwater input anomaly $\gamma_A$. The freshwater anomaly is weakening the AMOC, which results in a reduced meridional heat transport northward. This cooling effect in the northern region makes $\Delta T$ smaller. The northern temperature spans a range almost four times bigger than the one in the south in the safe overshoot scenario and nine times bigger in the unsafe overshoot one; hence the main contribution to $\Delta T$ comes

from the northern box.

However, $\Delta T$ has a smaller impact on $\Delta\rho$ compared to $\Delta S$. The reason is that thermal expansion coefficient $\alpha_T = 10^{-4}C^{\circ-1}$ is smaller than the haline contraction coefficient $\alpha_S = 7.6 \cdot 10^{-4}$ psu$^{-1}$ in the linear equation of state used in the model. So density changes are more affected by salinity changes than by temperature changes. In the safe overshoot scenario (solid lines), $\Delta T$ and $\Delta S$ hit their lowest value before rising again, eventually stabilizing at a positive value. Conversely,

in the unsafe overshoot scenario, both $\Delta S$ and $\Delta T$ continue to decline. In the collapsed state, the northern regions become less saline and cooler than the southern regions.

### 3.3 Physics of Safe/Unsafe Overshoot

As most of density changes occur in the northern box, we apply the salt balance Equation (7) to the northern box that extends over the Atlantic basin from $40°N$ to $60°N$. The values of the fluxes in the northern region (Figure 5a) show that, unlike in

the whole Atlantic basin case, there is a net freshwater input ($\Phi^s < 0$) through the surface and a net saline water input through lateral boundaries. When $\gamma_A$ starts to decrease, the response of the northern box reveals the precise mechanism of the AMOC safe/unsafe overshoot. In the safe overshoot scenario, a faster decline in freshwater forcing allows the lateral salt transport to surpass the effect of the freshwater input. This results in the change of sign of the tencency of the integrated salt content, from





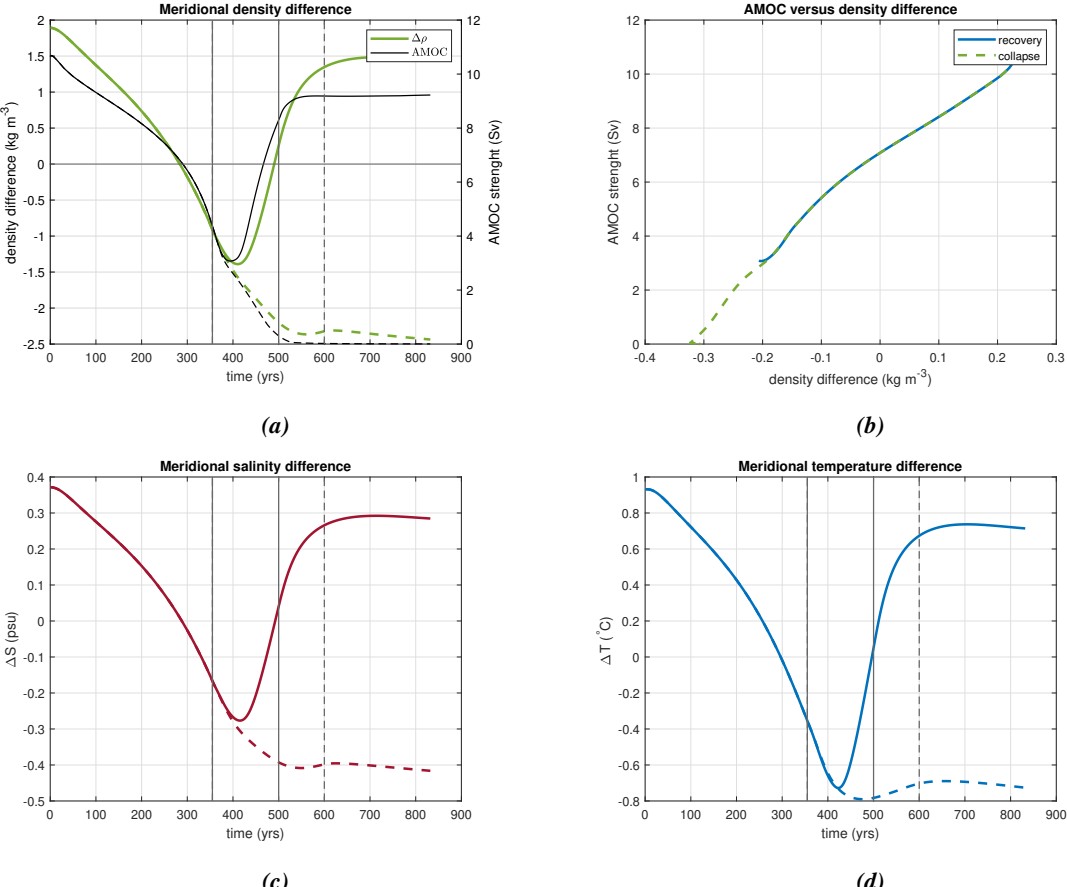

**Figure 4.** Diagnostics of changes in the northern and southern box associated with case A (Figure 2a-b). (a) The meridional density difference and the AMOC strength are plotted versus time. (b) The relation between $\Delta\rho$ and the AMOC strength. (c) Box averaged meridional salinity and (d) temperature differences. The solid curves are for the safe overshoot scenario, the dashed ones for the unsafe overshoot scenario. The vertical lines mark the points when the forcing changes.





negative to positive. Consequently, the northern box experiences an increase in salinity, reinforcing the AMOC and promoting
its recovery. In contrast, in the unsafe overshoot scenario, the freshwater forcing remains dominant over the lateral salt transport,
leading to a persistent net loss in salt storage $\left(\frac{d}{dt}\int_{northbox} SdV < 0\right)$. This results in a further freshening of the northern box
and in a consequent decrease of $\Delta S$ and $\Delta \rho$, which keeps weakening the AMOC and drives it toward a collapse. A detailed
examination of the lateral fluxes within the northern box, shown in Figure 5b, reveals that the advective salt transport through
its southern boundary of this box is the dominant flux. This advective flux, which transports saline waters northwards, plays a
crucial role in compensating the surface freshwater perturbation and thereby influences the AMOC behaviour under different
forcing scenarios.

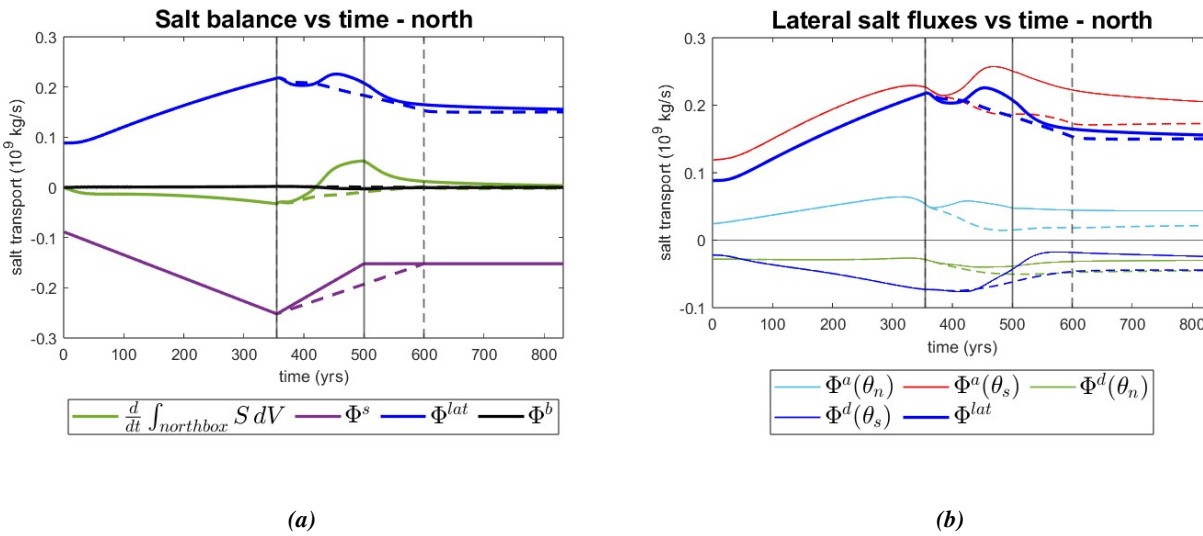

*(a)*                    *(b)*

**Figure 5.** (a) Terms in the integrated salt balance and (b) Lateral salt fluxes over the northern box along the AMOC trajectories in Figure 2a-b.
In both panels (a) and (b) , the solid curves are for the recovery case, the dashed ones for the collapse. The vertical lines mark the points
when the forcing changes. In the legend of panel (b), $\theta_s = 40°$N and $\theta_n = 60°$N.

Salt balance computations were similarly performed for the southern box but the corresponding plots have been omitted,
as the salt fluxes in the South Atlantic have a negligible impact on the AMOC recovery dynamics. In the South Atlantic, the
range of the freshwater anomaly is approximately an order of magnitude lower — about 20 times less — than that computed
in the North Atlantic, making its influence on the recovery or collapse minimal. Moreover, the lateral salt fluxes exhibit
a smaller range as well, being roughly 11 times smaller in the AMOC collapse scenario and 4 times smaller in the safe
overshoot scenario, in comparison to their North Atlantic counterparts. Therefore, the essence of the mechanism can be found
in the interplay between lateral salt transport and freshwater input in the northern box. It is the relatively early overcoming
of freshwater surface flux by lateral salt transport that characterizes the recovery phase of the AMOC. This is quantitatively
reflected in the shift from a negative to a positive time derivative of the integrated salt content, a change that signals the key





transition in the dynamics. The Atlantic basin, particularly its northern regions, begins to retain more salt, thereby restoring the meridional salinity (and density) gradients crucial for maintaining the AMOC.

The same analysis done for case A was applied to the two other cases (B and C) presented in Section 3.1 and the same mechanism is responsible for causing the safe/unsafe overshoot behaviour. In case B, the initial rate of increase in the freshwater

forcing is more rapid in the recovery scenario and the maximum value is reached earlier since the forcing peaks at the same magnitude in both scenarios ($0.235\,\mathrm{Sv}$). The faster forcing induces a larger overshoot, resulting in a stronger AMOC response as the freshwater input begins to decrease. The different AMOC values at the forcing peak (above $4\,\mathrm{Sv}$ for the safe overshoot and below $4\,\mathrm{Sv}$ for the unsafe overshoot case, respectively) significantly influence the lateral salt fluxes. Higher AMOC strength values enhance the salt transport into the northern box, while lower values correspond to weaker salt transport. As the forcing

declines, both scenarios experience the same surface forcing. As in case A, the key is that a stronger AMOC drives stronger lateral fluxes (mainly advective) in the northern box that are able to balance out and eventually surpass the freshwater input, making the time derivative of the salt content in the northern region change sign.

In case C, the freshwater input increases at identical rates but reaches different peak values at different times. This yields different AMOC strengths when the forcing peaks. A lower forcing (safe overshoot scenario) leads to a smaller overshoot and

hence corresponds to a stronger AMOC, which in turn generates more salt transport northward. This is enough for the lateral salt flux in the northern box to exceed the freshwater input and consequently makes the region more saline beneficial to restore the AMOC. Even though the decline of the forcing is slower, the time derivative of the integrated salt content in the north still switches sign early enough to lead to an AMOC recovery. On the other hand, in the unsafe overshoot scenario, the higher peak of the forcing corresponds to an already weakened AMOC. The advective salt fluxes alone in this case are unable to carry

enough saline water to the North Atlantic to surpass the freshwater forcing soon enough to make the region saltier.

## 4   Mathematical analysis of limiting cases

The aim of this section is to study the transient solutions of AMOC behavior analytically using a reduced mathematical model. The dynamics of the AMOC strength, indicated by $x$, can be approximated near a saddle-node bifurcation (which we know exists in the global ocean model used in section 3) by the following (Li et al., 2019) non-autonomous ordinary differential

equation (ODE):

$$\frac{dx}{d\tau} = -x^2 - f(\tau) \qquad (11)$$

where the forcing is a piece-wise linear function:

$$f(\tau) = \begin{cases} -\alpha(1-\tau) & \tau_0 < \tau < \tau_1 \\ \beta(1-\tau) + k & \tau > \tau_1 \end{cases} \qquad (12)$$

with $\alpha, \beta > 0$ and $k = (\alpha + \beta)(\tau_1 - 1)$ to make the forcing continuous.

The advantage of considering the above one-dimensional model is that analytical solutions can be determined. Using this framework, we will be able to determine a priori the rate of decline in forcing — given a fixed peak and rate of increase — that

 

allows the AMOC to achieve a safe overshoot. Finally, we will compare these analytical findings with the numerical results obtained from the global ocean model.

### 4.1 Local form near the saddle-node bifurcation

Our starting ODE is the general form of the saddle-node bifurcation

$$\frac{dX}{dt} = aX^2 + bX + c - \gamma(t). \tag{13}$$

where $\gamma(t)$ is defined as in Equation (4). In order to get the equation in the form of Equation (11), a rescaling is needed. We thereby apply the following change of variables

$$\begin{cases} x = -at_{tip}\left(X + \frac{b}{2a}\right) \\ \tau = \frac{t}{t_{tip}} \\ \alpha = -am_1 t_{tip}^3, \qquad \beta = -am_2 t_{tip}^3. \end{cases} \tag{14}$$

with $t_{tip} = \frac{c}{m_1} - \frac{b^2}{4am_1}$. Rewriting Equation (13) as

$$\frac{dX}{dt} = a\left(X + \frac{b}{2a}\right)^2 + \left(c - \frac{b^2}{4a}\right) - \gamma(t),$$

we can substitute the new variables and get:

for $t \in [t_0, t_1]$

$$\frac{dx}{d\tau} = \frac{dx}{dt}\frac{dt}{d\tau} = t_{tip}\frac{dx}{dt} = t_{tip}\left(-a \cdot t_{tip}\right)\frac{dX}{dt} = -at_{tip}^2\frac{dX}{dt}$$

$$= -x^2 + \alpha(1 - \tau)$$

for $t > t_1$, similar calculations lead to

$$\frac{dx}{d\tau} = -x^2 - \beta(1 - \tau) - k$$

In this way, the problem is in the form of Equation (11).

### 275  4.2  Analytical solution

Analytical solutions to Equation (11) can be found in terms of Airy functions (Li et al., 2019). We first compute the bifurcation diagram of the corresponding autonomous system assuming $\tau$ as bifurcation parameter (instead of time). The steady states are given by

$$x^\pm(\tau) = \pm\sqrt{-f(\tau)} \tag{15}$$





or more explicitly $x_1^+(\tau) = \sqrt{\alpha(1-\tau)}, x_1^-(\tau) = -\sqrt{\alpha(1-\tau)}, x_2^+(\tau) = \sqrt{-\beta(1-\tau)-k}, x_2^-(\tau) = -\sqrt{-\beta(1-\tau)-k}$. The functions $x_i^\pm(\tau), i = 1, 2$ form two parabolas which represent the bifurcation diagram. When a full solution of Equation (11) crosses one of these curves, its derivative becomes zero, which means that $x_i^\pm(\tau), i = 1, 2$ separate regions in the phase plane $(t, x)$ where the derivative has different signs.

Next, we compute the complete analytical solutions. Equation (11) is a Riccati equation, thus it can be transformed (Li et al., 2019) from a first order nonlinear ODE to a second order linear ODE through the change of variable $x = \frac{1}{u}\frac{du}{d\tau}$. In this way it becomes:

$$
\frac{du}{d\tau} = -f(\tau)u = -\begin{cases} -\alpha(1-\tau)u & \tau_0 < \tau < \tau_1 \\ (\beta(1-\tau)+k)u & \tau > \tau_1 \end{cases} \tag{16}
$$

We are interested in the solutions satisfying the initial condition $x(\tau_0) = x_0$ and, without loss of generality, we can take $\frac{du}{d\tau} = x_0$ and $u(\tau_0) = 1$. The solution to Equation (16) can then be expressed in terms of Airy functions ($Ai$ and $Bi$):

$$
u(\tau) = \begin{cases} \left[C_1 Ai(z) + C_2 Bi(z)\right]_{z=\sqrt[3]{\alpha}(1-\tau)} & \tau_0 < \tau < \tau_1 \\ \left[K_1 Ai(w) + K_2 Bi(w)\right]_{w=\sqrt[3]{\beta}(\tau-1)-k/\beta^{2/3}} & \tau > \tau_1 \end{cases} \tag{17}
$$

Rescaling back in terms of $x$, the solution to Equation (11) is

$$
x(\tau) = \begin{cases} -\sqrt[3]{\alpha}\left[\dfrac{C_1 \frac{dAi(z)}{dz} + C_2 \frac{dBi(z)}{dz}}{C_1 Ai(z) + C_2 Bi(z)}\right]_{z=\sqrt[3]{\alpha}(1-\tau)} & \tau_0 < \tau < \tau_1 \\ \sqrt[3]{\beta}\left[\dfrac{K_1 \frac{dAi(w)}{dw} + K_2 \frac{dBi(w)}{dw}}{K_1 Ai(w) + K_2 Bi(w)}\right]_{w=\sqrt[3]{\beta}(\tau-1)-k/\beta^{2/3}} & \tau > \tau_1 \end{cases} \tag{18}
$$

where the constants are

$$
C_1 = \left[\frac{\frac{dBi(z)}{dz} + \frac{x_0}{\sqrt[3]{\alpha}}Bi(z)}{Ai(z)\frac{dBi(z)}{dz} - Bi(z)\frac{dAi(z)}{dz}}\right]_{z=\sqrt[3]{\alpha}(1-\tau_0)}
$$

$$
C_2 = \left[\frac{-\frac{x_0}{\sqrt[3]{\alpha}}Ai(z) - \frac{dAi(z)}{dz}}{Ai(z)\frac{dBi(z)}{dz} - Bi(z)\frac{dAi(z)}{dz}}\right]_{z=\sqrt[3]{\alpha}(1-\tau_0)}
$$

and

$$
K_1 = \left[\frac{\frac{dBi(z)}{dz} - \frac{x_0}{\sqrt[3]{\beta}}Bi(z)}{Ai(z)\frac{dBi(z)}{dz} - Bi(z)\frac{dAi(z)}{dz}}\right]_{w=\sqrt[3]{\beta}(\tau_1-1)-k/\beta^{2/3}}
$$

$$
K_2 = \left[\frac{\frac{x_0}{\sqrt[3]{\beta}}Ai(z) - \frac{dAi(z)}{dz}}{Ai(z)\frac{dBi(z)}{dz} - Bi(z)\frac{dAi(z)}{dz}}\right]_{w=\sqrt[3]{\beta}(\tau_1-1)-k/\beta^{2/3}}
$$

Figure 6a shows two examples of forcing that differ in the value of $\beta$ and Figure 6b shows the associated solutions. Continuous solutions as the one in blue reproduce the recovery of the AMOC, while the solutions with a vertical asymptote (in





red) are the collapses. The bifurcation diagram for both cases (treating $\tau$ as parameter) is also shown as the thin curves. As the analytical approximation is tailored for a neighborhood of a single saddle-node bifurcation, it excludes the off state of the AMOC. Consequently, the post-collapse behaviour cannot be captured within this analytical framework.

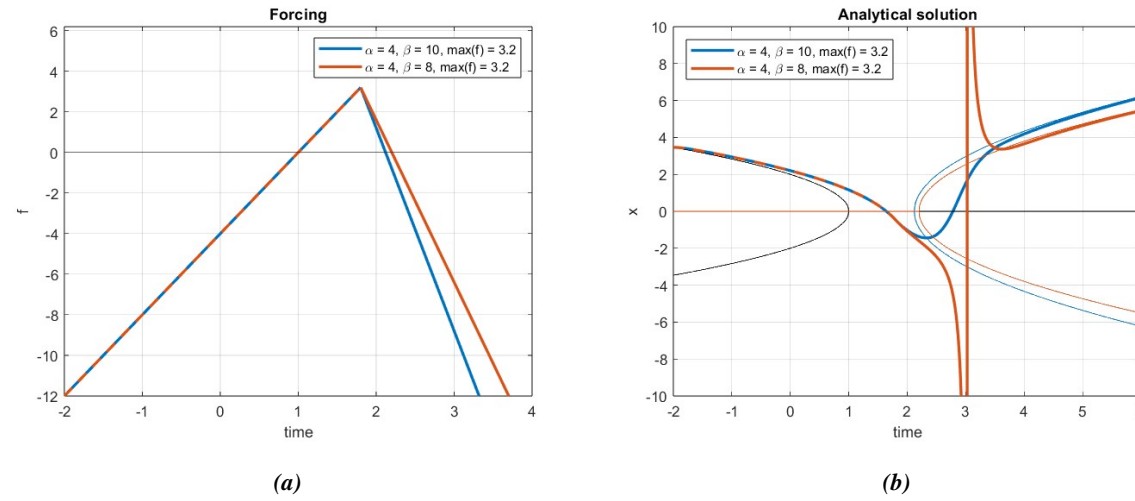

**Figure 6.** (a) Piece-wise linear forcing that increases for $\tau < \tau_1 = 1.8$ and decreases for $\tau > \tau_1$. The rate of decrease is different in the two cases: $\beta = 10$ for the blue curve, $\beta = 8$ for the red one, while $\alpha = 4$ for both cases. (b) Analytical solution of Equation (11) given the forcing in (a). The blue curve is a safe overshoot, the red one an unsafe one.

### 4.3 Conditions for a safe overshoot

Given that the main scenarios analyzed in Section 3 are the ones of case A, we focus here on studying conditions for a safe overshoot in the case of a fixed value of $\alpha$. We pick $\alpha = 84$ for reasons that will become clear in Section 4.4 below. First of all, the solution in the time interval with an increasing forcing ($\tau \in [\tau_0, \tau_1]$) has a collapse time at $\tau = \tau^*$, meaning that the solution reaches $-\infty$. The value of $\tau^*$ depends on the parameter $\alpha$ and it is given by the asymptote of the pullback attractor, which can be explicitly computed: $\tau^*(\alpha) = 1 + 2.338\alpha^{-1/3}$ (Li et al., 2019). This scenario corresponds to a prolonged increasing forcing that makes the AMOC collapse before the forcing starts decreasing. To avoid this kind of unsafe overshoot, we need $\tau^* \notin [\tau_0, \tau_1]$, i.e.

$$\tau_1 < \tau^*(\alpha). \tag{19}$$

From now on, we make sure that the condition in Equation (19) is satisfied and the solution is continuous on the interval $[\tau_0, \tau_1]$. For $\alpha = 84$, this means that the forcing must have its peak before $\tau = 1.5453$.

Solutions for different values of $\tau_1$ and $\beta$ are computed and a fraction of those is shown in Figure 7a-b. For the values of $(\tau_1, \beta)$ for which a collapse happens, the collapse time $\tau^*$ is computed (where $\tau = \tau^*$ is the equation of the asymptote). For each value of $\tau_1$, Figure 7c displays the collapse times $\tau^*$ as a function of the rate of decline $\beta$ of the forcing. As anticipated,





a smaller value of $\beta$ corresponds to a smaller $\tau^*$. This indicates that as the forcing decreases more rapidly, the collapse is postponed, and if the decline is sufficiently fast, the collapse can be entirely avoided.

Finally, for each $\tau_1$ we can take the largest value of $\beta$, say $\tilde{\beta}$, for which the solution exhibits a collapse. For all $\beta < \tilde{\beta}$, the forcing is decreasing in a slower way and hence all the associated solutions collapse. On the other hand, for all $\beta > \tilde{\beta}$, the solutions recover. Thus, the curve $(\tau_1, \tilde{\beta}(\tau_1))$, plotted in Figure 7d for $\alpha = 84$, divides the parameter space into safe and unsafe overshoots regions. The value of $\tilde{\beta}$ decreases as $\tau_1$ decreases. In fact, given that $\alpha$ is fixed, a smaller $\tau_1$ corresponds to a lower peak of the forcing, which in turn leads to a smaller decline of the AMOC. This allows for a longer AMOC overshoot (hence

a slower decrease in the forcing) while still having a recovery. The analytic solutions provide a clear picture of the response to the AMOC to the freshwater forcing taking into account the rate of the forcing decrease $\beta$ and time $\tau_1$ when the forcing begins to decrease (in the same way one could take the peak value of the forcing, as was done in the analysis of case C in Section 3). Hence, within this simple analytical framework, a critical curve has been found defining stability regions, similar to that in the conceptual model used in Ritchie et al. (2021).

## 4.4   Connection to the global ocean model: case A

We now compare the results obtained from the simple ODE in Equation (11) with the numerical simulations obtained with the global ocean model for which results were presented in Section 3. A quadratic fit is made near the tipping point in the bifurcation diagram (Figure 1b) where $\Psi_A$ is computed as a function of the bifurcation parameter $\gamma_A$. Hence, the AMOC strength $\Psi_A = X$ is used as the primary variable in (13) and the fit to determine the constants $a, b, c$ yields $a = -0.0060, b =$

$0.0658$ and $c = 0.0056$. In order to have recovery and collapse simulations while staying as close as possible to the saddle-node bifurcation, results for four additional global ocean model simulations (labelled a-d) are shown in Figure 8a-b. They all have $m_1 = 6.1972 \cdot 10^{-4}$ Sv/year and $\gamma(t_1) = 0.22$ Sv (with $t_1 = 355$ years), while they differ for the value of $m_2 = \{8.7, 3.7, 2.3, 1.1\} \cdot 10^{-4}$ Sv/year. To be able to compare the simulations to the analytical solutions, we transform the simulation data $\Psi_A(t)$ and the simulation parameters using Equation (14). The parameters $m_1, t_1$ and $m_2$ translate respectively to $\alpha =$

$84.061$, $\tau_1 = 1.18$ $(\max(f) = f(\tau_1) = 15.13)$ and $\beta = \{118.71, 49.89, 31.58, 15.03\}$.

The transformed AMOC trajectories are plotted versus time in Figure 8c-f (dashed curves) together with the analytical solution of the ODE in Equation (11) computed with the same forcing parameters. The green part (sol A) corresponds to the solution in the time interval of forcing growth, the red part (sol B) to the solution in the time interval of forcing decline. Figure 8c shows that the simulation with the largest $\beta$ (118.71) and hence the fastest decrease is the best approximated

simulation. The smaller $\beta$ gets, the worse the approximation becomes. In Figure 8d, for instance, the minimum of the analytical solution is larger (in absolute value) and is reached with a certain delay compared to the numerical simulation from the global ocean model. Figure 8e shows an AMOC recovery in the ocean model whereas the corresponding analytical solution of the ODE instead exhibits a collapse. In Figure 8f, both the numerical simulation and the analytical solution show an AMOC collapse. However, the simulation spends a longer time between the on and off state (it reaches the collapsed state at $\tau = 2.25$)

while the analytical solution goes to $-\infty$ much quicker (for $\tau^* = 1.65$).





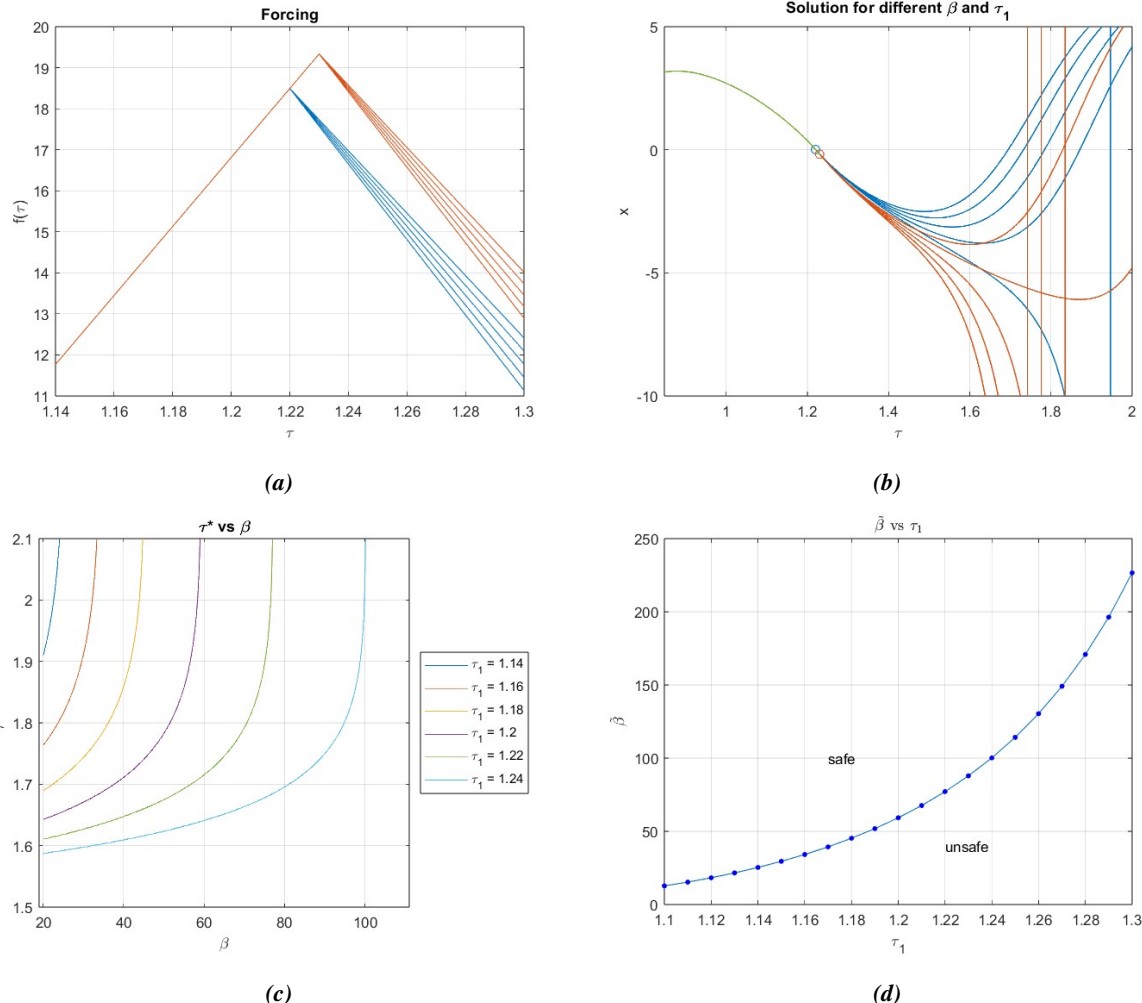

**Figure 7.** (a) Forcing and (b) associated solutions to Equation (11). The values of the parameters are: $\alpha = 84$, $\tau_1 = 1.22$ (in red), $\tau_1 = 1.23$ (in blue), $\beta \in [92, 76]$. (c) Collapse time $\tau^*$ vs decline rate $\beta$ of the forcing $f(\tau)$ for $\alpha = 84$ and different values of $\tau_1$ between 1.14 and 1.24. (d) The maximum value of $\beta$ ($\tilde{\beta}$) for which the solution collapses vs the time $\tau_1$ at which the forcing reaches its peak. This critical curve separates the safe and unsafe overshoot regions in the parameter space $(\tau_1, \tilde{\beta})$ for $\alpha = 84$.



One of the possible reasons why the approximation may not always work well is that the ODE approximates the behaviour of the AMOC only locally around the saddle-node bifurcation and it is possible that the simulations performed were not close enough to the tipping point. Another reason could be the choice of $\Psi_A$ as main variable $X$. The state variable in the global ocean model is multi-dimensional, while the bifurcation diagram used for the quadratic fitting is just a one-dimensional projection.

Although the strength of the AMOC does exhibit a saddle-node bifurcation, it is also part of the projection of the state variable onto the eigenvector associated with this bifurcation. Incorporating the Lyapunov-Schmidt reduction method (Guckenheimer and Holmes, 1990) could help in addressing this issue. This technique reduces the complexity of the multi-dimensional problem by decomposing it into simpler, lower-dimensional form while capturing the essence of the bifurcation behaviour. This method would allow to identify the critical eigenvector that governs the bifurcation and to obtain a more accurate fit of the behavior of

the global ocean model near the bifurcation point, but its application is outside the scope of this paper.

## 5 Summary and Discussion

We investigated the behaviour of the Atlantic Meridional Overturning Circulation (AMOC) in response to overshoot scenarios under freshwater forcing. Our findings are in line with previous research that has investigated AMOC tipping behaviour and the potential for recovery following temporary threshold exceedance (Ritchie et al., 2021). Slow-onset tipping elements like

the AMOC may exhibit a delayed collapse, allowing for carefully managed overshoot scenarios that avoid irreversible state transitions. Our results expand on these studies by identifying the precise physical mechanisms for a safe or unsafe overshoot of the AMOC in a more detailed global ocean model.

We apply a piecewise linear freshwater forcing, that grows to a maximum above the tipping point and then decreases towards a constant value. Our results confirm that overshoots can be safe under certain scenarios even if the AMOC exceeds the tipping

point, contributing to the understanding of transient tipping behavior within the climate system. Similarly to Ritchie et al. (2021), our results show that the AMOC response is highly sensitive to both the rate at which the freshwater forcing increases to its peak and then decreases, as well as the initial strength of the AMOC before the forcing begins to decline. Specifically, we find that faster declines in forcing after reaching a peak are more likely to enable safe overshoot trajectories, allowing the AMOC to eventually recover. Conversely, prolonged exposure to high freshwater forcing — which mainly causes the AMOC

to be weaker — significantly increases the risk of a transition to a collapsed state, which does not recover once the freshwater forcing settles to a constant value.

The key to understand the different behaviour of the AMOC under different forcings is to be found in the salt fluxes of the northern North Atlantic region. When the lateral salt fluxes that transport saline water northward can counterbalance the freshwater input, making the region effectively saltier, the AMOC is able to recover, otherwise it is driven toward collapse. This

delicate interplay between surface and advective salt fluxes presents a tangible metric that could serve as a reliable indicator of the AMOC trajectory towards recovery. Specifically, the time derivative of the integrated salt content in the northern region goes through zero and changes sign when the AMOC is headed toward a recovery. The North Atlantic starts to receive more saline water, reinstating a larger meridional salinity difference which directly affects the meridional density difference that





**Figure 8.** Global ocean model simulations considered for the comparison with the analytical solutions. (a) Forcing scenario and (b) the associated AMOC trajectories with the corresponding colors. The purple forcing has the slowest rate of decrease and its AMOC trajectory shows a collapse; the other simulations show an AMOC recovery. (c-f) Four rescaled AMOC trajectories of the global ocean model (dashed lines) and the analytical solutions computed with the corresponding forcing parameters.





drives the AMOC. Although the global ocean model has a very course resolution and many other major limitations (Weijer
et al., 2003; Dijkstra, 2007), this mechanism is expected to be robust.

Indeed, this understanding could be useful in guiding global climate policies to mitigate and avoid the long-term conse-
quences of an AMOC collapse (Armstrong-McKay et al., 2022). For instance, similarly to what is done for the last analysed
scenario (case C in Section 3), one could combine different forcing shapes, rates, and peaks to explore a broader range of
AMOC responses. By varying these factors, one could identify threshold conditions under which the AMOC recovers or
collapses. Testing slower forcing declines in recovery scenarios or sharper declines in collapse scenarios may reveal clear
transition zones, that could help create a response map or define a scaling law, with forcing rates and peaks as parameters, to
systematically map the AMOC responses across various scenarios. Such an approach would extend the strictly local (to the
saddle-node) approach of Ritchie et al. (2019), who established an inverse-square law between time spent by the AMOC over
the tipping point and amplitude of the overshoot.

In this context, the analytical approximation in Section 4 could serve as an initial step towards refining predictions of the
AMOC responses to different forcing parameters. This framework opens the way to identify regions of safe/unsafe overshoot
delimited by critical curves within the parameter space (Li et al., 2019). It could provide a clearer picture of the AMOC
sensitivity to external forcing overshoot utilizing solely the bifurcation diagram from the global ocean model. However, it
is crucial to acknowledge its limitations in capturing the complex dynamics of the ocean. While the approximation serves
as a valuable conceptual tool, at the moment its simplicity comes at the cost of detailed accuracy. The model locality and
reduced complexity does not account for any feedback mechanisms, spatial heterogeneities and other nonlinear processes that
are critical in the real-world behaviour of the AMOC. Future work could aim to refine this analytical framework to enhance
its accuracy while still maintaining a level of simplicity that allows for broader accessibility and application in the context of
AMOC studies.

To enhance our understanding of the potential for AMOC recovery in transient overshoot scenarios, future research could
integrate the insights gained from our study into state-of-the-art climate models. Notably, the CESM model, for which the
AMOC tipping point has been detected through quasi-equilibrium simulations (van Westen et al., 2024), provides a promising
framework for such investigations. Simulating controlled freshwater forcings that mimic realistic overshoot trajectories would
involve an increase in freshwater input faster than those used to identify the AMOC threshold. After having reached a forcing
peak beyond the detected tipping point, the rate of freshwater decrease should be strategically adjusted to explore conditions
leading to both recovery and collapse. This approach would allow detailed study of the salt transport terms in a more detailed
model, offering critical insights into the processes governing AMOC stability and resilience.

The implications of our findings extend beyond theoretical interest, offering insights relevant to climate policy. The urgency
to understand and minimize climate tipping risks has been recognized in international climate policy for the first time at the 27th
Conference of the Parties (COP27). The Paris Agreement was aiming to limit the global temperature increase to $1.5°C$ above
pre-industrial levels (UNFCCC, 2015). However, recent data show that global warming has already reached $1.2°C$ (Forster
et al., 2024), and current climate policy scenarios are estimated to result in $2.6°C$ warming above pre-industrial levels (Rogelj
et al., 2023) by the end of this century (with a range of $1.7 - 3.0°C$). Even if the global mean temperature was to be stabilized





below $1.5°$C in the long term, a temporary overshoot above $1.5°$C is a clear possibility (Bustamante et al., 2023), underlining
the urgency that potential impacts and associated risks of such an overshoot need to be assessed.

Finally, the AMOC plays a vital role in regulating Northern Hemisphere climate and a permanent AMOC collapse would
likely have severe impacts on global weather patterns, potentially leading to altered precipitation and more extreme climate
events (van Westen et al., 2024; Armstrong-McKay et al., 2022; Meccia et al., 2023). Our study suggests that a controlled,
temporary overshoot in freshwater forcing — analogous to transient emissions scenarios — could provide policymakers with
a degree of flexibility in carbon emission targets, provided the decline in forcing is managed to support AMOC recovery.

*Code and data availability.* The model data and Matlab scripts used to generate the plots will be made available upon acceptance of the
manuscript.

*Author contributions.* AFR and HAD conceptualized the study. AFR acquired the results. Both authors contributed to writing the manuscript.

*Competing interests.* The authors declare that they have no conflict of interest.

*Financial support.* This research has been supported by the European Research Council through the ERC-AdG project TAOC (PI: Dijkstra,
project 101055096).



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
