# Peer review of "Physical characterization of the boundary separating safe and unsafe AMOC overshoot behaviour"

_EGUsphere, 2025_

## Author Comment (AC1)

**MS-No.:** egusphere-2025-45

**Title:** Physical characterization of the boundary separating safe and unsafe AMOC overshoot behaviour

**Authors:** Aurora Faure Ragani and Henk A. Dijkstra

**Point-by-point reply to reviewer #2**

April 16, 2025

We thank the reviewer for their careful reading and for the useful comments on the manuscript. The comments of the reviewer are in italic font below and our response is in normal font.

**Overview**

*The paper studies the recovery or its failure around an AMOC breakdown for an intermediate complexity model depending on a freshwater forcing parameter. It has been found before that for certain values of the parameter the model possesses a branch of steady states whose stable large amplitude region is identified with the AMOC. The branch undergoes a fold bifurcation that is identified as an AMOC tipping point. In this paper the parameter is chosen piecewise linear in time such that it overshoots the fold value for an intermediate time interval. Numerical computations are presented for several choices of the time parameterisation showing "safe" scenarios that give recovery of the AMOC, as well as "unsafe" ones, where the recovery fails. The results are interpreted physically, suggesting that the lateral salinity advection can distinguish these. Finally, a comparison with analytical results for a scalar fold point are presented. I think the topic is timely and of broad interest, fitting to Earth System Dynamics. However, I have a few comments that should be addressed in a revision.*

**Author's reply:**
Thank you for your positive evaluation of the paper.

**Main comments:**

1. *In order to shorten the manuscript, I recommend to revise section 4. On the one hand, (13) and the discussion on how to achieve the normal*

*form is not needed, and in fact misleading as noted in the next point. On the other hand, I do not think section 4.3 is needed as the formulae appear to be directly taken from Li et al 2019.*

**Author's reply:**
The analysis is indeed based on the results of Li et al. (2019), but the new element is the piecewise continuous forcing, so the formulae are not directly taken from that paper.

**Changes in manuscript:**
To improve readability, we will move many of the details of the analytical analysis in section 4.3 to an appendix.

2. *In section 4.4 it seems the authors use $\Psi_A$ as an ad hoc variable $X$ and suggest this relates to a closed ODE (13). However, such a relation requires a center manifold reduction and the resulting coefficients can strongly differ from those obtained here. Hence, I would strongly recommend to do this, and I do not see much added difficulty: one just needs a kernel eigenvector of the linearisation at the fold point, and the corresponding adjoint kernel eigenvector. I do not understand what the authors later mean by Lyapunov-Schmidt reduction.*

**Author's reply:**
We addressed this issue already in section 4.4 of the original paper as a reason why the results of the analytic model and the numerical model differ. It is outside the scope of this paper to compute the center manifold reduction for such a complicated ($\sim$ 500,000 degrees of freedom) model as one also needs also to determine the nonlinear terms. The Lyapunov-Schmidt method is very similar to the center manifold reduction method (but the reduction is algebraic instead of dynamic). In fact, in many finite-dimensional cases, they produce equivalent reduced equations (especially for stationary bifurcations).

**Changes in manuscript:**
We will explain in more detail the reason for the mismatch of the numerical and analytical results and also mention the center manifold reduction method in section 4.3.

**Minor/typo:**

We thank the reviewer for pointing out the corrections; we will follow all suggestions in the revised paper.

1. *Abstract: I wonder about the definite formulation "[The AMOC] is an important tipping element". In line 16 this is rather differently only "potential".*

   **Author's reply:**
   We will remove 'potential' in line 16.

2. *l5: What is meant by the term "fully-implicit (global ocean model)"? I think it would deserve a brief explanation.*

   **Author's reply:**
   Agreed. We will include a short explanation.

3. *In section 3.1: Is there some significance of the times in years mentioned here?*

   **Author's reply:**
   No, the time scale just serves as illustration.

4. *l65: Explain why the cos(theta) part is removed from $F_S$ (also elsewhere, e.g. in (8), (9)).*

   **Author's reply:**
   The cos(theta) part comes from integrating over a spherical surface.

5. *Fig 1b: The upper part of the dashed line branch looks a bit edgy. Is this getting smoother with smaller continuation stepsize?*

   **Author's reply:**
   Indeed, but this is not crucial here so not recomputed.

6. *l95: I think there is a typo: the parameter in the third time interval in (4) should be time-independent.*

   **Author's reply:**
   No, there is no typo. After $t_2$, the forcing is constant.

7. *l384: course should be coarse*

   **Author's reply:**
   Well spotted and this will be corrected.

---

## Author Comment (AC2)

**MS-No.:** egusphere-2025-45

**Title:** Physical characterization of the boundary separating safe and unsafe AMOC over-
shoot behaviour

**Authors:** Aurora Faure Ragani and Henk A. Dijkstra

**Point-by-point reply to reviewer #1**

April 16, 2025

We thank the reviewer for their careful reading and for the useful com-
ments on the manuscript. The issues raised by the reviewer are in italic font
and our responses are in normal font.

**Overview**

*This manuscript studies the safe and unsafe slowdown beyond a tipping
point of the AMOC. It suggests a characterization through the salinity bud-
get and the compensating role of the lateral salinity advection. Also the study
suggests a theoretical derivation to asses the possibility of safe and unsafe
AMOC overshoot. I do find this work timely and very interesting. This
work is well within the scope of Earth System Dynamics. After reading the
manuscript I have a few specific concerns. Also, the manuscript organization
can be significantly improved. Hence I recommend a major revision. Please
see below my major, specific, and minor comments.*

**Author's reply:**
Thank you for this (basically) positive evaluation.

**Major comments:**

1. *Model description: The forcings need to be better described. I under-
   stand/guess that you do not use seasonal cycle. Please say so. Also
   what is the forcing for temperature (heat flux I guess)? What is the
   strategy for wind forcing? How are they set? A figure of the forcings
   (and maybe its perturbation) won't harm...*
   *A summary figure of the reference states would be useful SST, SSS,
   Stratification, zonal average T and S in the Atl., AMOC. It is impor-
   tant for the reader to grasp the realism of the model. [These figures will*

*also help us have an intuitive feeling of the configuration, i.e., horizontal resolution.]*

*I appreciate that you do not want to repeat everything regarding model description. However a quick summary of the physics encompassed in this model seems key to fully understand the rest of the paper. What is taken into account in the momentum equation (geostrophy, nonlinear advection of momentum, hydrostatic, type of viscosity)? Same thing for T and S evolution (do you have Gent-McWilliams? is it a isopycnal/diapycnal diffusion?). Overall writing down the equations and quickly describing them would be useful.*

*I am not fully familiar with the type of numerical model you are using (i.e., "implicit" model). However you should clarify in the model section that this type of model can also been run in "classical" way: time integration allowing the computation of trajectories. This is where a set of equations (for both the model dynamics and the method to compute the steady states and their stability) can help the reader. Please clarify this point.*

**Author's reply:**
We will clarify the issues requested (e.g. fully-implicit model), but we are not going to include equations and add plots of basic features of solutions of the model as this has been done extensively elsewhere. The use of continuation methods is not essential in this paper (because we mainly present results of transient simulations) and hence do not need to be detailed here.

**Changes in manuscript:**
The model description will be extended to clarify specific issues, and references will be provided on the other issues raised, including the continuation methods.

2. *Freshwater forcing experiments:*
*I am a bit confused by (1). Does $\gamma_A = 0$ ensure Q=0? If not it means that you apply a correction on the surface flux computed for equilibrium (i.e., you set it to a 0 global mean). I would guess that it should disrupt the steady state. Is it the case? Does the steady state remain a steady state under $\gamma_A = 0$ and $Q \neq 0$? Even if somehow technical, this*

*question is a fundamental question regarding your experimental setup. If your starting point is not a steady state, your experiment setup is problematic. Please clarify this point in the manuscript.*

**Author's reply:**
Yes, the diagnosed surface freshwater flux from the Levitus restoring solution has $Q = 0$.

**Changes in manuscript:**
This will be mentioned explicitly in the revised manuscript.

3. *Too unneeded lengthy analyses: For the key message of the manuscript I find that a significant part of section 3 is not needed.*
*Overall the key message (i.e., following section 3.2 and section 4) is on the forcing slope of the recovery (i.e., m2 coefficient). I am not convinced that the other numerical experiments are so useful, especially since they are not discussed in the rest of the study. This makes the manuscript longer without a proper study of these other cases. I suggest to concentrate on Case A (and give the rationale for this choice) and remove Case B and C. The title and abstract should also then reflect that. Alternatively, a full discussion and theoretical analyses of Case B and C are needed (i.e, section 3.2 and section 4).*
*I find that most of the section 3.2 is not needed. The figure 5 is the key message. It carries everything you need for your argument and is easily introduced without the need for the other previous diagnostics. Since your forcing is localized in the subpolar north Atlantic testing the freshwater budget in this region makes perfect sense. (You could mention that you have tested other regions/locations, without giving the full analysis.)*
*Overall I find that the manuscript is describing "all" the experiments and analyses that the authors have made, but failed to organize it in a "simple" story. Removing the unnecessary parts should help clarify the key message. (Key message that I find extremely interesting and worth publishing.)*

**Author's reply:**
Indeed, case B and C are not needed for the understanding of the mechanism of the safe/unsafe overshoot, so we will only mention these cases in the discussion section. However, the details of the analysis in section 3.2 for case A are needed to understand the mechanism and hence will not be shortened.

**Changes in manuscript:**
Results for cases B and C, including Fig. 2c-f plus their discussion, will be removed.

4. *The quality of figures is really low...Please improve the quality of the figures and use thicker lines (and larger fonts) when possible for legibility.*

   **Author's reply:**
   Agreed.

   **Changes in manuscript:**
   The quality of all figures will be improved.

**Specific comments:**

1. *l.21-24: I do not see the link between these two sentences. Why would the dominance of the positive salinity feedback over the negative temperature feedback be linked to their type of forcing? For me and in an hand-waving way, the feedback are set by the meridional gradient of salinity and temperature (which are ultimately set by the amplitude of the forcing, but not his "type").*
   *This not to say that the type of forcing does not have an effect on the system. The temperature have a negative feedback related to surface restoring, whereas the salinity has not such second feedback. Maybe I am missing something, but in this case you need a longer explanation with some references. Please clarify these two sentences. l.66-69: Since you are studying the AMOC (i.e., Atlantic-MOC) a definition specific for the Atlantic basin (setting $\lambda_W$ and $\lambda_E$ for the Atlantic) would be helpful and would simplify the description of the next sentence.*

**Author's reply:**
This is rather well known; the different restoring time scales are actually essential in the Stommel (1961) box model.

**Changes in manuscript:**
We will explain this in more detail with reference to the Stommel (1961) box model.

2. *l.78: I strongly suggest to make the new zero branch visible!*

   **Author's reply:**
   Agreed.

   **Changes in manuscript:**
   The zero branch will be shown in Fig. 1b and the text in l.78 will be adapted.

3. *Section3.1: A short summary at the end of the experiments description and results would have been nice. It should express something like: "too slow increase or decrease as well as too high peak prevent the recovery". Whereas the two last points make intuitive sense, the former is apparently more paradoxical: pushing the system faster is actually safer! (Note that it is quite intuitive from a dynamical system point of view. You do not want the system to equilibrate when pushed over the tipping point.)*

   **Author's reply:**
   Good suggestion.

   **Changes in manuscript:**
   A short summary will be provided.

4. *Initial experiments (Fig.1): This figure is illustrative but quite useless for the rest of the study. I suggest you to consider using the trend used in the exp B. This will probably already show what you mean (influence of the forcing rate change), but be more consistent with the rest of*

*the study. If you want to make a point that slow forcing "follows" the branch, you can used an extremely slow increase to make your point (quasi-autonomous system). But I understand that it is not an argument you are making.*

**Author's reply:**
Fig. 1 serves to show the connection between the time-dependent behavior and the underlying bifurcation diagram and hence is important for the rest of the paper.

**Changes in manuscript:**
No changes in the manuscript.

5. *In all figures showing the stability branches: I suggest to use different line types for stable and unstable state. Also please show both stable branch.*

**Author's reply:**
Suggestion followed.

**Changes in manuscript:**
Figures will be adapted.

6. *Choice of $m_1$ and $m_2$: I wonder if the timescale is realistic. Compared to CMIP6 model simulations this is quite long.*

**Author's reply:**
The time scales can be compared by the input due to melting of the Greenland Ice Sheet (which is not in CMIP6 models), but this is not essential for explaining the mechanism of the safe/unsafe overshoot.

**Changes in manuscript:**
We will explicitly mention the time scales implied by $m_1$ and $m_2$.

7. *l.113-116: Do you mean the forcing spend more time over the "threshold". Because I do not understand how you assess how long the AMOC spend over a tipping point... Following this idea. I think it would be*

*nice to plot the AMOC as a function of time and assess the time spend with an AMOC value over the AMOC($\gamma_A$) at steady state. We might learn something comparing this time and the time spend with a forcing over the threshold. Also the timing and the difference in timing (of AMOC and forcing over their respective threshold) might be insightful.*

**Author's reply:**
Because we know the position of the saddle-node bifurcation, we can exactly determine the time spend over the tipping threshold. This can be directly calculated from the value of $\gamma_A$ and the rate of the forcing.

**Changes in manuscript:**
No changes in the manuscript.

8. *Case C: I don't think this is a good design since now you are modifying two parameters: the peak and the recovery pace. Since the latter have been shown to have an influence (exp A), I would advice you to use the same recovery as blue, so a single parameter is tested.*

**Author's reply:**
Based on the earlier comments of the reviewer, results for case C will no longer be presented.

**Changes in manuscript:**
No further changes in the manuscript.

9. *Fig.2: Overall the blue curve should be identical (which is not, they are cut for c-d and e-f). This means that you have only 4 different experiments. With a good color choice for the curves, I am quite convinced that you can put them all on a single panel!*
*I won't be against looking at AMOC as a function of time... Maybe over the first column panels or as a third column inserted in the center. I don't think that the first vertical line is put at the correct location. Also I don't understand why the negative values is not visible... Finally the branch should show stable and unstable part.*
*Why not running the model for long enough to properly see the $t > t_2$ phase. This seems important. In particular it seems to do something*

*surprising in (b) with a running away of $\Psi$ for the constant value of $\gamma_A$. This behavior should be clarified and tested for other scenarios.*

**Author's reply:**
Based on the earlier comments of the reviewer, results for case C will no longer be presented.

**Changes in manuscript:**
Figure 2 will be modified and only panels a) and b) will remain.

10. *Section3.2: The last part (box region) is extremely descriptive. I don't think that we learn anything that could not be summarized in a single sentence (i.e., dominance of salinity) or is almost obvious (i.e., change in the North, where the forcing is).*

**Author's reply:**
We do not agree. This analysis is required to understand the mechanisms in section 3.3 and is far from trivial. It is also not descriptive as we support the arguments made with detailed quantitative results.

**Changes in manuscript:**
No changes in text.

11. *l.151: If I understand correctly here the S stands for longitude-depth section. This is a new terminology that is not defined. Introducing new terminologies that are not needed is potentially confusing (especially by using S that is also used for Salinity). Simply mentioned that you introduce the total oceanic salinity flux divergence this encompassed both advection and diffusion. Maybe a notation with $\Delta\Phi$ would be more intuitive.*

**Author's reply:**
The use of $S$ can indeed be confusing.

**Changes in manuscript:**

We will use a different symbol ($\mathcal{S}$) to indicate the section surface.

12. *l.158: I do not expect anything... Maybe if you show an AMOC time series I would have expected something... But a decrease of AMOC, is not necessarily related to a change in freshwater flux divergence if the slow-down is consistent between latitudes. Please clarify.*

    **Author's reply:**
    Agreed.

    **Changes in manuscript:**
    This will be better explained.

13. *l.160: I don't think so. It is under-compensating. The flux divergence should decrease faster to keep equilibrium (dS/dt=0).*

    **Author's reply:**
    No, we do not agree.

    **Changes in manuscript:**
    Text will be clarified to avoid confusion.

14. *l.170-171: It would be interesting to mention that from the ocean salt flux divergence only the flux at the southern boundary change sign between the two experiments (and seem to control the flux divergence). This important difference between dashed and solid red lines of Fig.3b seems to be controlling the change of $\Phi_{lat}$.*

    **Author's reply:**
    Thank you for this suggestion.

    **Changes in manuscript:**
    This will be mentioned in the revised manuscript.

15. *l.72-173 – Southern box definition: What is the rationale for that? Is the result strongly sensitive to that? It seems more natural to use a tropical and polar box. The tropical box should go from 35S to 40N. Would that dramatically change the results?*

    **Author's reply:**
    It has been shown in earlier work that the meridional density between such a southern box and northern box best correlate with the AMOC strength.

    **Changes in manuscript:**
    The motivation for the choice of the southern box will be extended, including the appropriate references.

16. *l.196-198: This does not make any sense, simply (always) plot $\alpha\Delta T$ and $\beta\Delta S$.*

    **Author's reply:**
    That is indeed better.

    **Changes in manuscript:**
    Figure 4c and 4d will be adapted.

17. *Fig.4: I am confused by the panel b compare to a. In (a) for value of $\Delta\rho$ values ranging from -0.3 to 0.3 kg m-3, we do not have value of the AMOC from 0 to 12 Sv but from 6 to 7 Sv. Something is inconsistent here. This does not bring confidence on the rest of the analysis...*

    **Author's reply:**
    Well spotted! There was indeed an error in the axis labels. It has no consequence for the rest of the analysis, which was correct.

    **Changes in manuscript:**
    Figure 4b will be corrected.

18. *Fig.5: Overall I feel that it is the only useful figure of the analysis. The rest should be removed, to focus on the result.*

    **Author's reply:**
    We do not agree. Overall, we will not follow the suggestions of the reviewer to (i) remove most of the (novel) analysis and (ii) to include many (old) material on model formulation and basic solutions.

    **Changes in manuscript:**
    No changes in the text.

19. *l.228-245: If you want to keep case B and C (which I am not sure to be a wise choice), you should have a figure summarizing the result: something equivalent to Fig. 5 which seems to be the (only) key one for case A.*

    **Author's reply:**
    We will delete cases B and C, except for a short description in the discussion section.

    **Changes in manuscript:**
    No further changes in text.

20. *Section 4.1: I do not get the use of a new equation... What is the point? Why (13)? What are the parameters? Any references?*

    **Author's reply:**
    This is just the general form of a saddle-node bifurcation, which is needed here to determine the local steady solution structure of the global ocean model near tipping.

    **Changes in manuscript:**
    We will explain the need for this equation better with an appropriate reference.

21. *Fig.6: I understand that the solutions are the thick lines. What are the thin lines? What are the meaning of black, red, and blue in the thin lines? Asymptotic results? (Why the horizontal X=0 line changes color from red to black?) This should be described in the caption and explicitly linked to the derivation (i.e., equation number).*

    **Author's reply:**
    The thin lines were explained in l. 301.

    **Changes in manuscript:**
    We will better explain all the curves in a revised caption to Figure 6.

22. *Section 4.4: My feeling is that it should be a dedicated section connecting the numerical and the theoretical results.*

    **Author's reply:**
    That feeling is correct.

    **Changes in manuscript:**
    No changes needed.

23. *l.353: $\Psi_A$ is quite non-linear. Have you tried with the subpolar average salinity ?*

    **Author's reply:**
    We have indicated how a better scalar variable can be determined, but it is outside the scope of the paper to perform this analysis.

    **Changes in manuscript:**
    No changes needed.

**Minor comments:**
We thank the reviewer for pointing out these suggestions for corrections and we will follow all of these in the revision.

1. *l.21: Damping timescale is probably not the correct term for salinity forcing.*

   **Author's reply:**
   Restoring time scale is more often used and we will adapt this.

2. *l.69-70: "$\Psi$ in the Atlantic" does not make sense, since $\Psi$ is a zonal average... You might want to say "$\Psi$ for the Atlantic" or more precisely "$\Psi$ restricted to the Atlantic sector (by setting $\lambda_W$ and $\lambda_E$ appropriately".*

   **Author's reply:**
   Suggestion will be followed.

3. *l.77: $\gamma_A- = 0.054$ – Out of curiosity, have you check if it corresponds to a changing sign of the FOV at the Atlantic entry? (I am quite critical of this hypothesis that has been used as an established-theoretical-result in the field....)*

   **Author's reply:**
   Yes, we did and will provide a reference.

4. *l.78: Is it 0.0522 or 0.054 as you just mentioned "0.054" on l.77? Please clarify the value at the bifurcation. Also make sure that this is correctly displayed in all figures.*

   **Author's reply:**
   It is 0.054 and will be corrected. Thanks for spotting.

5. *Fig1-caption: It won't harm clarifying that $\bar{\gamma}_A$ is the value at the bifurcation in (b).*

**Author's reply:**
Suggestion will be followed.

6. *l.99: You mean a linear increase? Please say so and give a reference.*

   **Author's reply:**
   This statement was indeed unclear and will be corrected.

7. *l.111 and elsewhere: Since the 90's, Sv year$^{-1}$ is preferred to Sv/year.*

   **Author's reply:**
   Agreed and will be corrected.

8. *Figure 3 and others: You should put minus sign when appropriate, so that the we can visually sum the curve to obtain the oceanic salt flux divergence.*

   **Author's reply:**
   This is easy also with the current plots.

9. *l.165 "this time": not defined.*

   **Author's reply:**
   Indeed! Will be clarified.

10. *Figure3-caption: replace "case" by "of case A"*

    **Author's reply:**
    Suggestion will be followed.

11. *l.188-191: 3 sentences saying the same thing...*

**Author's reply:**
Thanks. Will be shortened.

12. *l.189-190: verb? l.193: replace "bigger than" by "as big as"*

    **Author's reply:**
    Suggestion will be followed.

13. *l.194: replace "bigger" by "as big as"*

    **Author's reply:**
    Suggestion will be followed.

14. *l.219: replace "less" by "as less as"*

    **Author's reply:**
    Suggestion will be followed.

15. *l.221: replace the two "smaller" by "as small as"*

    **Author's reply:**
    Suggestion will be followed.

16. *References: A few DOIs seem wrong...*

    **Author's reply:**
    Will be corrected.